# Robotic Single-Site Plus Two-Port Myomectomy versus Conventional Robotic Multi-Port Myomectomy: A Propensity Score Matching Analysis

**DOI:** 10.3390/jpm12060928

**Published:** 2022-06-03

**Authors:** Seyeon Won, Su Hyeon Choi, Nara Lee, So Hyun Shim, Mi Kyoung Kim, Mi-La Kim, Yong Wook Jung, Bo Seong Yun, Seok Ju Seong

**Affiliations:** 1Department of Obstetrics and Gynecology, CHA Gangnam Medical Center, CHA University School of Medicine, Seoul 06135, Korea; djtong85@naver.com (S.W.); k345@chamc.co.kr (S.H.C.); naradd@naver.com (N.L.); simuso@chamc.co.kr (S.H.S.); ra13811@chamc.co.kr (M.K.K.); mila76@cha.ac.kr (M.-L.K.); sunghunpapa@naver.com (Y.W.J.); 2Department of Obstetrics and Gynecology, CHA Ilsan Medical Center, CHA University School of Medicine, Goyang 10414, Korea; bosungyun@hanmail.net

**Keywords:** uterine myomectomy, robotic surgical procedures, laparoscopy, uterine fibroids

## Abstract

Background: Robotic single-site plus two port myomectomy (RSTM) was designed to reduce the number of incision sites while retaining the advantage of conventional robotic multi-port myomectomy (CRM). This study aimed to explicate RSTM and compare surgical outcomes between it and CRM. Methods: The medical records of 146 patients who had undergone RSTM and 173 who had undergone CRM were reviewed. The surgical outcomes between them were compared by propensity score matching (PSM) analysis. Results: The PSM analysis showed no statistically significant inter-group differences in patient characteristics. With regard to surgical outcomes, the RSTM group enjoyed shorter operative time (148.30 ± 44.8 vs. 162.3 ± 47.4 min, *p* = 0.011), less hemoglobin decrement (1.8 ± 0.9 vs. 2.3 ± 1.0 g/dL, *p* < 0.001), and shorter duration of hospital stay (5.4 ± 0.7 vs. 5.8 ± 0.7 days, *p* < 0.001). Conclusions: RSTM was associated with shorter operative time relative to CRM. Further prospective studies are needed in order to more fully investigate the advantages of RSTM.

## 1. Introduction

One of the newest technical improvements in minimally invasive surgery has been the introduction of the da Vinci^®^ robotic surgical platform (Intuitive Surgical, Inc., Sunnyvale, CA, USA) [1]. This robotic system provides improved ergonomics, wristed instrumentation with increased freedom of movement, hand-tremor buffering, enhanced depth perception, and a three-dimensional field of view [2].

In 2004, Advincula et al. reported the first conventional robotic multi-port myomectomy (CRM) [3], and three later studies detailed favorable surgical outcomes [4,5,6]. For CRM, four incision sites, including the 12-mm camera port at the umbilicus, two 8-mm side ports, and a 12-mm or 5-mm laparoscopic assistant port, are required. Robotic single-site myomectomy (RSSM) has been introduced to minimize surgical injury by reducing the number of ports; however, many studies have reported problematic issues due to inherent technical problems, such as limited traction with semi-rigid instruments, as well as range-of-motion limitations [7,8].

Robotic single-site plus two-port myomectomy (RSTM) was designed to reduce the number of incision sites while retaining the advantage of CRM. The RSTM apparatus includes a 23-mm multi-channel single port at the umbilicus and just two 8-mm side ports, without CRM’s additional assistant port. Through the multi-channel single port at the umbilicus, the 12-mm robotic camera and the laparoscopic assistant instrument are inserted.

We hypothesized that, with the RSTM system, the operative time would be shorter thanks to the easier morcellation due to the larger umbilical incision. Thus, in the present study, we compared surgical outcomes between RSTM and CRM and performed propensity score matching (PSM) analysis to ensure inter-group comparability.

## 2. Materials and Methods

A retrospective cohort study with the approval of the relevant institutional review boards (GCI-2022-01-005) was conducted in a single gynecological surgery center using data collected between September 2020 and October 2021. The medical records of 146 patients who had undergone RSTM and 173 who had undergone CRM, both procedures by five expert surgeons, were reviewed. Patient data, including age, body mass index (BMI), marital status, parity, previous surgical history, and myoma features, were extracted from the records. The preoperative hemoglobin level was checked within 3 months of surgery, and postoperative hemoglobin was recorded on the first postoperative day. Complications were defined as when there was ileus, fever, or wound dehiscence within 30 days from surgery.

### 2.1. Surgical Methods

#### 2.1.1. RSTM

RSTM by the da Vinci Si or Xi robotic system (Intuitive Surgical, Inc., Sunnyvale, CA, USA) was performed. A 23-mm incision established the single-site port of entry at the umbilicus. Subsequently, the glove port (Nelis, Seoul, Korea), followed by the 12-mm camera port (at the umbilicus) and the two 8-mm side ports were inserted (Figure 1A and Figure 2). The remaining port site of a glove port was used for assistant laparoscopic instruments. Once all of the trocars had been inserted, the surgical cart was docked vertically. Monopolar curved scissors were wielded in the right arm, and bipolar forceps in the left. A diluted vasopressin solution (0.25 U/mL concentration) was infused into the myoma. The monopolar curved scissors in the right robotic arm made the incision, while the forceps in the left arm applied counter-traction. V-loc™ (Covidien, Dublin, Ireland) was utilized for multiple-layer suturing of the uterine wall, and in-bag scalpel morcellation was performed to retrieve the myomas through the umbilical incision.

#### 2.1.2. CRM

CRM was performed using the da Vinci Si or Xi robotic system. The 12-mm camera port at the umbilicus, the two 8-mm side ports, and the 12-mm assistant port were inserted (Figure 1B). The subsequent procedures were the same as in RSTM, except for the morcellation method: that is, in CRM, the myomas were retrieved by electric power morcellation.

### 2.2. Statistical Analysis

Student’s *t*-test for comparison of continuous variables and the χ^2^ test for categorical variables were used. For determination of nonparametric statistics, Fisher’s exact test was used. 1:n PSM with a nearest-neighbor matching algorithm was performed to minimize selection bias. The proportion of women with peritoneal adhesion, proportion of women with history of abdominal surgery, and tumor weight were selected as variables for propensity matching, because those variables were statistically different between groups. The analyses were performed using SPSS version 24.0 (IBM Inc., Armonk, NY, USA), and *p*-values < 0.05 were considered statistically significant.

## 3. Results

### 3.1. Baseline Characteristics

The patients’ baseline characteristics are provided in Table 1. In the CRM group relative to RSTM, heavier tumor weight (204.0 ± 147.2 vs. 172.5 ± 128.5 g, *p* = 0.044), lower proportion of women with history of previous abdominal surgery (13.9 vs. 24.7%, *p* = 0.014), and lower proportion of women with peritoneal adhesion (13.9 vs. 23.3%, *p* = 0.030) were found. Otherwise, there were no significant inter-group differences. Table 2 shows the post-PSM baseline characteristics. As is apparent, there were no significant inter-group differences.

### 3.2. Surgical Outcomes

Table 3 summarizes the surgical outcomes. The RSTM group relative to CRM enjoyed shorter operative time (150.0 ± 46.2 vs. 163.6 ± 48.5 min, *p* = 0.011), less hemoglobin decrement (1.8 ± 0.9 vs. 2.3 ± 1.0 g/dL, <0.001), and shorter duration of hospital stay (5.4 ± 0.7 vs. 5.8 ± 0.7 days, *p* < 0.001). After the PSM, a similar tendency was observed. Again, the RSTM group had a shorter operative time (148.30 ± 44.8 vs. 162.3 ± 47.4 vs. min, *p* = 0.011), less hemoglobin decrement (1.8 ± 0.9 vs. 2.3 ± 1.0 g/dL, *p* < 0.001), and a shorter duration of hospital stay (5.4 ± 0.7 vs. 5.8 ± 0.7 days, *p* < 0.001). Although less hemoglobin decrement (1.8 ± 0.9 vs. 2.3 ± 1.0 g/dL, <0.001) was found in the RSTM group, neither estimated blood loss (213.2 ± 221.4 vs. 226.0 ± 182.7 min, *p* = 0.587) nor proportion of women receiving transfusion (7.2 vs. 3.0%, *p* = 0.091) were statistically different between the groups. There was no significant difference regarding the occurrence of postoperative complications either (*p* = 0.604). Two patients in the RSTM group and one in the CRM group experienced postoperative paralytic ileus. With conservative management, bowel function was restored. One febrile complication and one case of wound dehiscence occurred in both groups. There was no intraoperative complication and conversion to laparotomy in either group.

## 4. Discussion

To the best of our knowledge, this is the first cohort study to have evaluated surgical outcomes of RSTM versus those of CRM. Although there have been two studies on RSTM [8,9], there is none that has weighed the pros and cons of RSTM against CRM’s.

One of the most interesting aspects of our findings is the shorter operative time of RSTM compared with that of CRM. We considered the following three possible reasons. First, with RSTM, omitting the morcellation process is possible for small myomas, which is to say that, thanks to the 23-mm diameter of the umbilical trocar, myomas of smaller diameter can be extracted directly, and quickly and simply, even in the docking state. In cases of multiple myomas, myomas have to be strung together with every single enucleation so as not to lose a myoma prior to initiation of morcellation [10]. However, again, for small myomas of less than 23-mm diameter, no such time-consuming stringing procedure is necessary. Second, in-bag scalpel morcellation was performed in RSTM, while, in CRM, electric power morcellation was the method utilized. There are much conflicting data on the effectiveness of electric power morcellation [11,12], and controversy persists, particularly with regard to whether it is superior to in-bag morcellation. Sanderson et al. reported that manual morcellation was 21 min faster than electric morcellation [12]. On the other hand, Zullo et al.’s meta-analysis indicated that morcellation operative time was slightly longer for in-bag manual morcellation than for electric power morcellation (mean difference 2.59 min, 95% CI 0.45 to 4.72), though, due to the low quality of their data, they could not be certain of the respective methods’ effects [13]. Notwithstanding the lack of consensus on which morcellation method is faster, we believe that it is highly likely that RSTM’s shorter operative time in the present study was owed to in-bag morcellation. Third, RSTM’s shorter operative time may have been due to the simplified procedure for suturing or suture removal in the RSTM system. In the CRM system, this is carried out through the 12-mm assistant port. However, sometimes suture material becomes obstructed in the middle of the port, or misses the trocar entrance, thus prolonging an operation. In the RSTM system, contrastingly, suture material can be transferred through the umbilical trocar without having to adjust or undock or eliminate robotic instruments.

Although less hemoglobin decrement (1.8 ± 0.9 vs. 2.3 ± 1.0 g/dL, *p* < 0.001) was found in the RSTM group, we do not ascribe any significance to this, because the preoperative hemoglobin level had been checked when the operation was scheduled, which was within 3 months of surgery. In addition, more patients received a transfusion (7.2 vs. 3.0%, *p* = 0.091) in the RSTM group than in the CRM group, though the inter-group difference was not statistically significant.

We also certainly expect that RSTM could influence postoperative recovery as relates to hospital stay and pain alleviation due to the reduced number of incision sites; however, we acknowledge that our results concerning the shorter hospital days of RSTM cases carry no objective significance, as patients, according to our center practice, can easily extend their period of stay if they so desire.

The major concern with RSTM has been the limited motion of the assistant instrument due to its extracorporeal interaction with the robotic camera. Actually, in early cases, some collisions occurred and caused inconvenience. However, technical difficulties are rarer with a well-trained assistant and/or employment of the da Vinci 30-degree camera. Another concern is the risk of incisional hernia. In Connell et al.’s meta-analysis [14], single-incision laparoscopic surgery, of which the RSTM system is a mode, was correlated with a three-fold increase in hernia probability relative to conventional laparoscopic surgery, due to the longer incision length at the umbilicus.

Nevertheless, the RSTM system has several advantages. First, because of the RSTM system’s longer umbilical incision, ‘in-bag scalpel morcellation’ can be more easily conducted. In-bag scalpel morcellation is a challenging process in the CRM system, contrastingly, because the umbilical incision is very small, about 12mm. In addition, due to the potential for dissemination of occult malignancy, use of power electric morcellation has been restricted by the U.S. Food and Drug Administration (FDA) [15]. At our center, over the past six years, there have been 13 cases of STUMP and 12 cases of uterine sarcoma among 2026 robotic myomectomies, though no case arose during our study period. The second advantage of RSTM is that suture materials and small myomas can be removed easily through the umbilical multi-channel trocar without having to adjust, undock, or eliminate robotic instrumentation. Third, due to the above-noted first and second advantages, operative time could be shortened. Fourth, enhanced cosmetic satisfaction on the part of patients can be anticipated.

Our study does have limitations, however, due specifically to its retrospective nature. First, operative time was not evaluated in detail. If docking and morcellation time were measured and compared, more meaningful results could have been drawn. Second, surgical method selection was not randomized; even the morcellation method differed between the groups. Therefore, even though PSM was implemented, selection bias could have affected our results. Third, neither cosmetic outcomes nor degree of postoperative pain relating to incision-site number were assessed. Whereas the RSTM system does entail fewer incision sites, umbilical incision length is 1 cm longer than in the CRM system. Long-term complications, such as incidence of incisional hernia, were not evaluated either.

In conclusion, this study is the first to have assessed surgical outcomes of RSTM versus those of CRM. RSTM, having fewer incision sites, was associated with shorter operative time. Possible as-yet-unrevealed advantages of RSTM due to its reduced incision-site number should be investigated in further prospective studies.

## Figures and Tables

**Figure 1 jpm-12-00928-f001:**
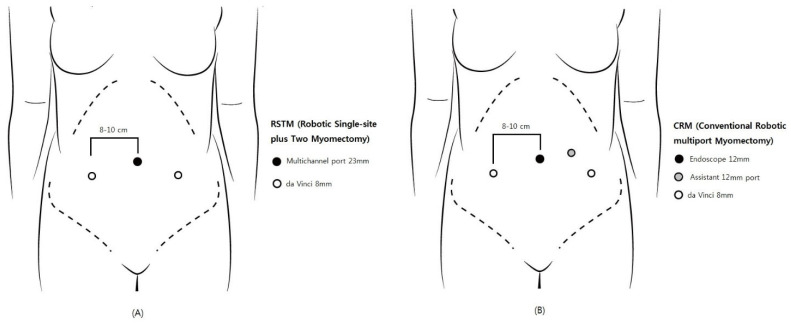
Comparison of trocar placement: (**A**) robotic single-site plus two-port myomectomy (RSTM); (**B**) conventional robotic multi-port myomectomy (CRM).

**Figure 2 jpm-12-00928-f002:**
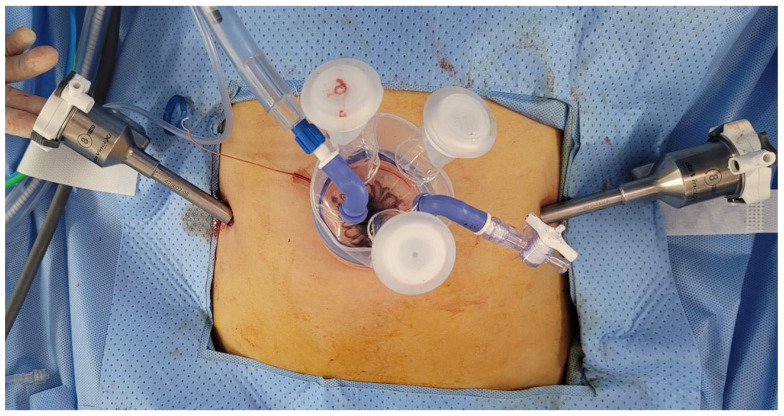
Actual trocar placement for robotic single-site plus two-port myomectomy (RSTM).

**Table 1 jpm-12-00928-t001:** Baseline characteristics of myomectomy patients.

Characteristics	RSTM (*n* = 146)	CRM (*n* = 173)	*p*
Age, years	38.0 ± 5.3	38.4 ± 5.4	0.445
BMI, kg/m^2^	22.6 ±3.3	23.2 ± 3.3	0.126
Nulliparous			0.746
No	23 (15.8)	25 (14.5)	
Yes	123 (84.2)	148 (85.5)	
Previous abdominal surgery			0.014
No	110 (75.3)	149 (86.1)	
Yes	36 (24.7)	24 (13.9)	
Peritoneal adhesion			0.030
No	112 (76.7)	149 (86.1)	
Yes	34 (23.3)	24 (13.9)	
Concurrent surgery			0.983
No	115 (78.8)	134 (77.5)	
Ovarian cystectomy	20 (13.7)	25 (14.5)	
USO	0 (0)	2 (1.2)	
Focal adenomyomectomy	8 (5.5)	11 (6.4)	
Salpingectomy	3 (2.1)	1 (0.6)	
Total myoma, n	7.0 ± 5.5	7.1 ± 5.3	0.607
Largest myoma			
Size, cm	6.8 ± 2.3	7.0 ± 2.5	0.270
Location			0.423
Anterior	53 (36.3)	72 (41.6)	
Posterior	72 (49.3)	63 (36.4)	
Fundal	13 (8.9)	22 (12.7)	
Anterior fundal	6 (4.1)	9 (5.2)	
Posterior fundal	2 (1.4)	7 (4.0)	
Type (FIGO classification)			0.529
Submucosal (1–2)	5 (3.4)	9 (5.2)	
Deep intramural (3–4)	60 (41.1)	66 (38.2)	
Intramural (5)	40 (27.4)	44 (25.4)	
Subserosal (6)	32 (21.9)	43 (24.9)	
Pedunculated (7)	3 (2.1)	7 (4.0)	
Intraligamentary (8)	6 (4.1)	4 (2.3)	
Tumor weight, g	172.5 ± 128.5	204.0 ± 147.2	0.044

Values are presented as number (%), median (range) or mean ± standard deviations. RSTM, robotic single-site plus two myomectomy; CRM, conventional robotic multiport myomectomy; BMI, body mass index; FIGO, International Federation of Gynecology and Obstetrics; USO, unilateral salpingo-oophorectomy.

**Table 2 jpm-12-00928-t002:** Baseline characteristics of myomectomy patients after propensity score matching (PSM).

Characteristics	RSTM (*n* = 125)	CRM (*n* = 169)	*p*
Age, years	37.8 ± 5.4	38.5 ± 5.4	0.294
BMI, kg/m^2^	2276 ± 3.2	23.2 ± 3.4	0.172
Nulliparous			0.925
No	18 (14.4)	25 (14.8)	
Yes	107 (85.6)	144 (85.2)	
Previous abdominal surgery			0.264
No	102 (81.6)	146 (86.4)	
Yes	23 (18.4)	23 (13.6)	
Peritoneal adhesion			0.962
No	107 (85.6)	145 (85.8)	
Yes	18 (14.4)	24 (14.2)	
Concurrent surgery			0.819
No	102 (81.6)	132 (78.1)	
Ovarian cystectomy	14 (11.2)	24 (14.2)	
USO	0 (0)	2 (1.2)	
Focal adenomyomectomy	7 (5.6)	10 (5.9)	
Salpingectomy	2 (1.6)	1 (0.6)	
Total myoma, n	6.9 ± 5.2	7.1 ± 5.3	0.547
Largest myoma			
Size, cm	7.0 ± 2.3	6.9 ± 2.3	0.856
Location			0.463
Anterior	45 (36.0)	71 (42.0)	
Posterior	62 (49.6)	62 (36.7)	
Fundal	12 (9.6)	21 (12.4)	
Anterior fundal	5 (4.0)	8 (4.7)	
Posterior fundal	1 (0.8)	7 (4.1)	
Type (FIGO classification)			0.572
Submucosal (1–2)	3 (2.4)	9 (5.3)	
Deep intramural (3–4)	48 (38.4)	64 (37.9)	
Intramural (5)	35 (28.0)	44 (26.0)	
Subserosal (6)	30 (24.0)	41 (24.3)	
Pedunculated (7)	3 (2.4)	7 (4.1)	
Intraligamentary (8)	6 (4.8)	4 (2.4)	
Tumor weight, g	177.7 ± 122.0	197.3 ± 137.0	0.204

Values are presented as number (%), median (range), or mean ± standard deviations. RSTM, robotic single-site plus two myomectomy; CRM, conventional robotic multiport myomectomy; BMI, body mass index; FIGO, International Federation of Gynecology and Obstetrics; USO, unilateral salpingo-oophorectomy.

**Table 3 jpm-12-00928-t003:** Surgical outcomes and morbidity before and after propensity score matching (PSM).

Characteristics	Total Data			In PSM Data		
	RSTM (*n* = 146)	CRM (*n* = 173)	*p*	RSTM (*n* = 125)	CRM (*n* = 169)	*p*
Operative time, mins	150.0 ± 46.2	163.6 ± 48.5	0.011	148.3 ± 44.8	162.3 ± 47.4	0.011
EBL, mL	216.8 ± 228.9	230.4 ± 184.5	0.558	213.2 ± 221.4	226.0 ± 182.7	0.587
Hemoglobin decrement, g/dL	1.8 ± 0.9	2.3 ± 1.0	<0.001	1.8 ± 0.9	2.3 ± 1.0	<0.001
Transfusion			0.096			0.091
No	136 (93.2)	168 (97.1)		116 (92.8)	164 (97.0)	
Yes	10 (6.8)	5 (2.9)		9 (7.2)	5 (3.0)	
Hospital stay, days	5.4 ± 0.7	5.8 ± 0.7	<0.001	5.4 ± 0.7	5.8 ± 0.7	<0.001
Laparotomy conversion			>0.999			>0.999
No	146 (100.0)	173 (100)		125 (100.0)	169 (100)	
Yes	0 (0.0)	0(0.0)		0 (0.0)	0 (0.0)	
Complications			0.565			0.604
None	140 (97.2)	171 (98.8)		122 (97.6)	167 (98.8)	
Ileus	2 (1.4)	0		1 (0.8)	0	
Fever >3 days	1 (0.7)	1 (0.6)		1 (0.8)	1 (0.6)	
Wound dehiscence	1 (0.7)	1 (0.6)		1 (0.8)	1 (0.6)	

Values are presented as number (%), median (range), or mean ± standard deviations; RSTM, robotic single-site plus two myomectomy; CRM, conventional robotic multiport myomectomy.

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
