# Peer review of "Robotic Single-Site Plus Two-Port Myomectomy versus Conventional Robotic Multi-Port Myomectomy: A Propensity Score Matching Analysis"

_jpm, 2022, doi:10.3390/jpm12060928_

Round 1

Reviewer 1 Report

I read this article with interest. This study compared the surgical outcomes for robotic single-site plus two-port myomectomy (RSTM) and conventional robotic multi-port myomectomy (CRM). The study showed that RSTM was associated with shorter operative time than CRM. The results of this study may be used in advancing surgical care for our gynecology patients. The manuscript was presented well. I only have a few questions and suggestions to improve the manuscript further.

Abstract

Line 22Please remove “vs.” in this: 162.3±47.4 vs. min, p=0.011

Line 23 Please add “p” before <0.001 in this line: (1.8±0.9 vs. 2.3±1.0 g/dL, <0.001)

Introduction

Please discuss previous studies comparing RSSM with CRM. What are the practical advantages of this method compared to CRM?

Methods

What was the basis of the sample size of 146 for RSTM and 173 for CRM? Did the authors compute for this sample size?

What were the inclusion and exclusion criteria for the study?

Did the authors include all patients who underwent RSTM and CRM between September 2020 and October 2021?

Results

How would the larger tumor size in the CRM group compared to the RSTM group confound the results? Do the authors think this could have affected the outcomes such as operative time and hemoglobin decrement?

Discussion

Lines 131 – 133 are incorrect. Please check this previous study which compares RSTM with CRM https://doi.org/10.1186/s12893-021-01245-9. Please discuss and compare your results with theirs.

Reviewer 2 Report

Thank you for the opportunity to read this well presented manuscript on Robotic single-site plus 2 port myomectomy versus conventional robotic multi-port myomectomy. 

The authors have explained well the rational for the study, their methods, their results and strengths and limitations. 

I have a few minor comments

In the methods there was not mention of capturing complication data. Was a classification system or grading system used?

Were the complications on intra-operative ones or within 30 days?

Line 95 - I believe the 86.1% should read 24.7% according to the table 1.

A comment about the propensity score, what it includes and why it is used may be helpful to the non-researcher clinician.
